# Characterization, Comparative Analysis and Phylogenetic Implications of Mitogenomes of Fulgoridae (Hemiptera: Fulgoromorpha)

**DOI:** 10.3390/genes12081185

**Published:** 2021-07-30

**Authors:** Wenqian Wang, Huan Zhang, Jérôme Constant, Charles R. Bartlett, Daozheng Qin

**Affiliations:** 1Key Laboratory of Plant Protection Resources and Pest Management of the Ministry of Education, Entomological Museum, Northwest A&F University, Yangling, Xianyang 712100, China; wanghan19931112@163.com (W.W.); 18335481296@163.com (H.Z.); 2O.D. Phylogeny and Taxonomy, Entomology, Royal Belgian Institute of Natural Sciences, Vautier Street 29, 1000 Brussels, Belgium; jerome.constant@naturalsciences.be; 3Department of Entomology and Wildlife Ecology, University of Delaware, 250 Townsend Hall, 531 S. College Ave., Newark, DE 9716-2160, USA; bartlett@udel.edu

**Keywords:** Fulgoroidea, Fulgoridae, phylogeny, mitogenome, genomics

## Abstract

The complete mitogenomes of nine fulgorid species were sequenced and annotated to explore their mitogenome diversity and the phylogenetics of Fulgoridae. All species are from China and belong to five genera: *Dichoptera* Spinola, 1839 (*Dichoptera* sp.); *Neoalcathous* Wang and Huang, 1989 (*Neoalcathous huangshanana* Wang and Huang, 1989); *Limois* Stål, 1863 (*Limois* sp.); *Penthicodes* Blanchard, 1840 (*Penthicodes atomaria* (Weber, 1801), *Penthicodes caja* (Walker, 1851), *Penthicodes variegata* (Guérin-Méneville, 1829)); *Pyrops* Spinola, 1839 (*Pyrops clavatus* (Westwood, 1839), *Pyrops lathburii* (Kirby, 1818), *Pyrops spinolae* (Westwood, 1842)). The nine mitogenomes were 15,803 to 16,510 bp in length with 13 protein-coding genes (PCGs), 22 transfer RNA genes (tRNAs), 2 ribosomal RNA genes (rRNAs) and a control region (A + T-rich region). Combined with previously reported fulgorid mitogenomes, all PCGs initiate with either the standard start codon of ATN or the nonstandard GTG. The TAA codon was used for termination more often than the TAG codon and the incomplete T codon. The *nad1* and *nad4* genes varied in length within the same genus. A high percentage of F residues were found in the *nad4* and *nad5* genes of all fulgorid mitogenomes. The DHU stem of *trnV* was absent in the mitogenomes of all fulgorids sequenced except *Dichoptera* sp. Moreover, in most fulgorid mitogenomes, the *trnL2*, *trnR*, and *trnT* genes had an unpaired base in the aminoacyl stem and *trnS1* had an unpaired base in the anticodon stem. The similar tandem repeat regions of the control region were found in the same genus. Phylogenetic analyses were conducted based on 13 PCGs and two rRNA genes from 53 species of Fulgoroidea and seven outgroups. The Bayesian inference and maximum likelihood trees had a similar topological structure. The major results show that Fulgoroidea was divided into two groups: Delphacidae and ((Achilidae + (Lophopidae + (Issidae + (Flatidae + Ricaniidae)))) + Fulgoridae). Furthermore, the monophyly of Fulgoridae was robustly supported, and Aphaeninae was divided into Aphaenini and Pyropsini, which includes *Neoalcathous*, *Pyrops*, *Datua* Schmidt, 1911, and *Saiva* Distant, 1906. The genus *Limois* is recovered in the Aphaeninae, and the Limoisini needs further confirmation; *Dichoptera* sp. was the earliest branch in the Fulgoridae.

## 1. Introduction

The family Fulgoridae is one of the large groups in the superfamily Fulgoroidea (Hemiptera: Auchenorrhyncha), consisting of 142 genera and 774 species [1], distributed worldwide but mainly in pan-tropical regions [1,2]. Many fulgorid species are brilliantly colored with an elongate and often strangely shaped head process. Some produce cuticular waxes, comprised mostly of keto esters in a variety of forms [3], including plumes that can extend well beyond the length of the abdomen (e.g., in *Cerogenes auricoma* (Burmeister, 1835)) [4]. Some fulgorids are agricultural pests, such as *Lycorma delicatula* (White, 1845); both nymphs and adults may cause direct feeding damage and indirect damage to plants through sooty mold growth on excreted honeydew [5,6].

The family Fulgoridae requires further assessment to obtain a robust phylogenetic assessment and resultant classificatory system [7]. The monophyly of the Aphaeninae, the second-largest subfamily of Fulgoridae comprising 35 genera and 207 species distributed mostly in the Old World [1] remains doubtful [7]. Moreover, the genus *Pyrops* Spinola, a large and showy genus of 69 species of tropical Asia and the Indomalayan region, remains of *incertae sedis* at the subfamily level [8]. Furthermore, the placement and status of Dichopterinae remain controversial [9,10,11,12,13,14]. More evidence, including mitogenomes, are needed to address these problems in Fulgoridae.

Here, we presented the complete sequenced and annotated mitogenomes of nine fulgorid species. We compared these mitogenomes with those previously published among Fulgoroidea and combined available mitogenomes to explore the diversity of mitogenomes and phylogenetics of Fulgoroidea, with particular reference to Fulgoridae.

## 2. Materials and Methods

### 2.1. Sample Collection and DNA Extraction

Specimens of nine fulgorid species were collected from China. All specimens were preserved in 100% ethanol and stored at −20 °C in the Entomological Museum of the Northwest A&F University. After morphological identification, the thoracic muscle tissue was used to extract total genomic DNA using the DNeasy DNA Extraction kit (Qiagen). Species identifications were based on Melichar (1912) [15] for the genus *Dichoptera*, Wang et al. (2020) [16] for the genus *Limois*, Constant and Pham (2018) [17] for the genus *Neoalcathous*, Constant (2010) [18] for the genus *Penthicodes* and Nagai and Porion (1996) [19] for the genus *Pyrops*.

The collection data of the extracted specimens are as follows:

Specimens of *Dichoptera* sp. were collected by Lijia Wang from Hainan Province, Mt. Jianfengling in August 2020 (Figure 1A); *Limois* sp. were collected by Daozheng Qin from Shaanxi Province, Huoditang in August 2019 (Figure 1B); *Neoalcathous huangshanana* Wang and Huang, 1989, was collected by Deliang Xu from Guangdong Province, Mt. Nanling in August 2020 (Figure 1C); *Penthicodes atomaria* (Weber, 1801) were collected by Wenqian Wang from Hainan Province, Mt. Diaoluo in April 2018 (Figure 1D); *Penthicodes caja* (Walker, 1851) were collected by Wenqian Wang from Yunnan Province, Guanping in April 2019 (Figure 1E); *Penthicodes variegata* (Guérin-Méneville, 1829) were collected by Wenqian Wang from Yunnan Province, Mengla in April 2019 (Figure 1F); *Pyrops clavatus* (Westwood, 1839) were collected by Wenqian Wang from Guangxi Province, Debao, Hongfeng forest park in June 2018 (Figure 1G); *Pyrops lathburii* (Kirby, 1818) were collected by Na Zhang from Guangxi Province in July 2017 (Figure 1H); *Pyrops spinolae* (Westwood, 1842) were collected by Wenqian Wang from Guangxi Province, Pingxiang in June 2018 (Figure 1I).

### 2.2. Mitogenomes Sequence Analysis

The whole mitogenomes of the nine species were sequenced using next-generation sequencing (NGS) (Illumina NovaSeq platform with paired-ends of 2 × 150 bp). Quality triming and assembly of raw paired reads were performed by Geneious 11.0.2 (Biomatters, Auckland, New Zealand) with default parameters [20]. The mitochondrial genome of *L. delicatula* (EU909203), *Pyrops candelaria* (Linné, 1758) (FJ006724), *Aphaena* (*Callidepsa*) *amabilis* (Hope, 1843) (MN025522) and *Aphaena* (*Aphaena*) *discolor nigrotibiata* Schmidt, 1906 (MN025523) (Hemiptera: Fulgoridae) [21,22] were chosen as bait sequences. Geneious 11.0.2 was used for mitogenomes annotation with *L. delicatula*, *Py. candelaria*, *A*. (*C*.) *amabilis* and *A*. (*A*.) *discolor nigrotibiata* (Hemiptera: Fulgoridae) used as references. All 13 PCGs were identified by open reading frames and translated into amino acids under the invertebrate mitochondrial genetic code. MITOS WebServer was run to identify and predict secondary structures of 22 tRNAs [23]. The secondary structures of tRNAs were mapped by Adobe Illustrator CS5 manually. After aligning with other mitogenomes in Fulgoridae, the rRNA genes and control region were annotated by the boundary of the adjacent tRNA genes. The tandem repeats of the control region were identified by the tandem repeats finder online server [24]. CGView was used to produce the whole mitogenome map [25]. PhyloSuite was performed to analyze the base composition and relative synonymous codon usage (RSCU) [26]. The sliding window analysis of 200 bp in a step size of 20 bp was performed with DnaSP v 5.0 to estimate nucleotide diversity (Pi) base on 13 PCGs and 2 rRNA genes from fourteen fulgorid mitogenomes (nine in this paper and five previously reported). Then, the DnaSP v5 was used to calculate the nucleotide diversity (Pi) of each PCG and rRNA [27].

### 2.3. Phylogenetic Analysis

Phylogenetic analyses were performed based on 13 PCGs and two rRNA genes among 53 species in Fulgoroidea, including 27 species in Delphacidae, 3 species in Ricaniidae, 2 species in Issidae, 1 species in Flatidae, 1 species in Lophopidae, 5 species in Achilidae and 14 species in Fulgoridae. The outgroups included five species from Membracoidea and two species from Sternorrhyncha. All the mitogenomes used in this study can be searched from GenBank (Table 1). Each PCG gene was aligned individually in codon-alignment mode under MAFFT and the results were concatenated by PhyloSuite [26]. Codon alignment modes were aligned using the G-INS-i (accurate) strategy, and RNA sequences were aligned using the Q-INS-i strategy normal alignment mode. Ambiguous sites and gaps in the alignments were removed using Gblocks v0.91b [28]. The optimal nucleotide substitution models and partition strategies were selected by PartitionFinder2. The best-fitting model was used for each partition with the greedy search algorithm and Bayesian information criterion (BIC) [26]. Four different datasets were concatenated for analyses: PCG123R matrix (all three codon positions of PCGs and two rRNA genes); PCG123 matrix (all three codon positions of PCGs); PCG12R matrix (the first and second codon positions of PCGs and two rRNA genes); PCG12 matrix (the first and second codon positions of PCGs). The maximum likelihood (ML) and Bayesian inference (BI) were employed based on four datasets for phylogenetic analyses. The maximum likelihood analyses were conducted by IQ-TREE [29], under an ML + rapid bootstrap (BS) algorithm with 1000 replicates. The Bayesian analyses were performed by MrBayes 3.2.6 [30] with two simultaneous runs of four chains each. Markov Chain Monte Carlo (MCMC) sampling estimated the posterior distributions using the settings for 5 × 10^6^ MCMC generations, with a relative burn-in of 25%, and MCMC termination when the average standard deviation of split frequencies fell below 0.01. PhyloBayes MPI v1.5a was used for phylogenetic reconstruction of the four datasets based on the site-heterogeneous model CAT + GTR on CIPRES [31,32]. Two independent trees were searched and the analysis was terminated when the two runs reached convergence (maxdiff below 0.3 and minimum effective size above 50). The initial 25% of each MCMC chain run was discarded as burn-in and a consensus tree was generated from the remaining trees combined from two runs.

## 3. Results

### 3.1. Mitogenome Organization and Gene Content

The complete mitogenome sequence of *Dichoptera* sp. was 15,803 bp, *Limois* sp. was 15,957 bp, *N**. huangshanana* was 16,510 bp, *Pe**. atomaria* was 16,093 bp, *Pe**. caja* was 16,040 bp, *Pe. variegata* was 15,814 bp, *Py**. clavatus* was 16,054 bp, *Py**. lathburii* was 16,104 bp and *Py**. spinolae* was 16,028 bp in length, respectively (Figure 2). Variations in the length of mitogenomes may be owing to a variable number of repeats in the control regions. The mitogenomes of nine fulgorid species encode 37 genes (13 PCGs, two rRNAs, and 22 tRNAs) and a control region (A + T-rich region) (Figure 2; Appendix A). All nine sequences exhibited the uniform gene arrangement as other planthopper mitogenomes. The nine mitogenomes displayed a heavy AT nucleotide bias, with an A + T% range from 73.6 to 77.9%: 77.9% in *Dichoptera* sp., 77.6% in *Limois* sp., 77.3% in *N. huangshanana*, 77.8% in *Pe. atomaria*, 76.5% in *Pe. caja*, 76.7% in *Pe. variegata*, 75.3% in *Py. clavatus*, 74.1% in *Py. lathburii* and 73.6% in *Py. spinolae*. This is moderate compared to levels found in other planthopper species. After comparing these nine mitogenomes, the composition skew analysis showed that all nine fulgorid species present a positive AT skew and a negative GC skew in the whole mitogenome and had a lower A + T content in tRNAs than in rRNAs. The control region (Table 2) has a positive AT skew and a negative GC skew in all nine mitogenome sequences except *Pe*. *caja*, which shows a negative AT skew and GC skew. A conserved overlap of 7 bp between *atp8* and *atp6* was found in nine sequences (Figure 3).

### 3.2. Protein-Coding Genes and Relative Synonymous Codon Usage

The total size of the 13 PCGs of nine mitogenomes ranged between 10,929 and 10,959 bp. In all nine mitogenomes, PCGs showed negative AT skew and GC skew. The A + T content of the third codon position was much higher than of the first and the second positions. The third codon position had a positive AT skew and negative GC skew in all nine species except *Pe. atomaria*, which had a negative AT skew and GC skew (Table 2). Most PCGs started with the codon ATN, except for *atp6* and *nad1* in *Py. lathburii*, which initiated with the codon GTG (Appendix A). This had also been found in other Fulgoroidea mitogenomes [21,33]. Comparing the nine sequenced mitogenomes, the result showed that *atp6* terminated with an incomplete T codon; *nad2*, *nad6*, *cox1*, *cox2* and *atp8* ended with a complete TAA codon; *nad5* ended with a TAN codon except for *Pe. caja* and *Pe. variegata* using the incomplete stop codon T. Transcribed truncated stop codons might be converted to TAA by polyadenylation [34]. In all nine sequenced mitogenomes, the ATG codon was used more often for starting than other codons and the GTG codon was the least used (Appendix A). The TAA codon appeared more often than the TAG codon, and the incomplete T codon was the least used in Fulgoridae. After comparing those sequences, we found *nad1* and *nad4* varied in length among species in the same genus (Appendix A). A high percentage of F residues appeared in *nad4* and *nad5* in all nine mitogenome sequences near the start position in *nad4*, and both the start and end position in *nad5*.

The result of relative synonymous codon usage (RSCU) (Figure 4) of the nine fulgorids shows that the codon usage of all genes had a strong bias toward the nucleotides A and T, particularly the third codon positions. Leucine (Leu), Phenylalanine (Phe), Isoleucine (Ile), Methionine (Met) and Serine (Ser) were used most frequently in the nine fulgorid mitogenomes sequenced. In addition, the codons Pro (CCG) and Thr (ACG) were absent in *Dichoptera* sp. The codons Arg (CGC) and Ser1 (AGC) were absent in *Limois* sp. The codons Pro (CCG) and Arg (CGC) deficiency was found in *Pe. atomaria*. The codon Arg (CGC) was not observed in *Py. clavatus*.

### 3.3. Transfer and Ribosomal RNA Genes

A total of 22 tRNA genes and two rRNA genes were identified in the same relative genomic positions as those observed for the previously sequenced fulgorid genomes [21,22]. The tRNAs had a positive AT skew and GC skew in all nine fulgorid species (Table 2). The total lengths of 22 tRNAs were 1399 to 1428 bp and 1943 to 1977 bp in rRNA (Table 2). The *trnV* was with a reduced DHU arm in all nine species except *Dichoptera* sp. (Appendix A). All nine fulgorid species were missing the DHU stem of this *trnS1*. The TΨC of *trnE* arms was missing in *Dichoptera* sp. All tRNA genes of the nine fulgorid species were the highly conserved, structures of 7 bp in the aminoacyl stem, 7 bp in the anticodon loop, and 5 bp in the anticodon stem, while the length of the DHU and TΨC arms was variable. In addition, *trnT*, *trnR*, and *trnL2* had an unpaired base in the aminoacyl stem in all nine fulgorid mitogenomes except *Dichoptera* sp., whose *trnT* were without an unpaired base in the aminoacyl stem. The anticodon stem of *trnS1* had an unpaired base in *N. huangshanana*, *Pe. atomaria* and *Py. clavatus.* Six types of unmatched base pairs (G-U, U-U, A-A, G-A, A-C and U-C) were found in the nine mitogenomes. The rRNAs had a negative AT skew and positive GC skew in nine sequenced mitogenomes (Table 2).

The large RNA genes were located between *trnL1* and *trnV*, with the sizes ranging from 1210 to 1224 bp, and the small ribosomal RNA genes were located between *trnV* and the control region, with the sizes ranging from 730 to 756 bp. The rRNA genes showed a heavy AT nucleotide bias, with A + T content 77.3% in *Dichoptera* sp., 78.2% in *Limois* sp., 76.8% in *N. huangshanana*, 77.1% in *Pe. atomaria*, 77.1% in *Pe. caja*, 77.2% in *Pe. variegata*, 76.3% in *Py. clavatus*, 75% in *Py. lathburii* and 75.3% in *Py. spinolae*, which were similar to those found in other sequenced Fulgoridae (Figure 2; Appendix A, Table 2).

### 3.4. Control Region

The putative control region (A + T-rich region) was annotated at the conserved position between *rrnS* and *trnI* (Table 2). Except for *Pe. caja* with negative AT skew and GC skew, all nine fulgorid species showed positive AT skew and negative GC skew in the control region. In the nine sequenced mitogenomes, the size of this region was 1327 bp to 2082 bp, and the A + T content of this region (more than 80%) was one of the highest within the whole mitogenome sequence (Table 2). One tandem repeat region was found in the control region of *Py. lathburii*, *Py. clavatus*, *Py. spinolae*, *Pe. caja*, *Pe. variegata* and *Dichoptera* sp.; *Pe. atomaria*, *N. huangshanana* and *Limois* sp. had two tandem repeat regions. The repeat unit of “TGCAAAAAAA(A)” was found in *Py. lathburii*, *Py. clavatus*, *Py. spinolae* and *N. huangshanana.* The repeat unit of “TTGCAAAAAA(A)” was found in *Pe. caja* and *Pe. variegata*, while a similar repeat unit of “TTGCAAAAA(A/T)(A)” was found in *Pe. atomaria*; TT/GCAAAAAAA(A) as the second repeat unit was found in *N. huangshanana. Limois* sp. has the repeat unit “TCATAAAAAA(A)”. The same repeat unit “TTGCAAAAAA(A)” was also found in two species, *A.* (*C.*) *amabilis* and *A.* (*A.*) *discolor nigrotibiata* in the genus *Aphaena* Guérin-Méneville, 1834 [22]. Moreover, 1–3 Poly(A) or Poly(T) could be found in those fulgorid species (Figure 5). Interestingly, a Poly(A) was consistently located near the 3′-end of the control regions in all the sequenced Fulgoridae species except *Pe. caja*, *Pe. variegata* and *N. huangshanana* (Figure 5).

### 3.5. Nucleotide Diversity

The results of nucleotide diversity based on 13 PCGs and two rRNA genes from fourteen sequenced Fulgoridae species (Figure 6) show that nucleotide diversity values range from 0.136 (*rrnL* and *rrnS*) to 0.264 (*atp8*). Comparing each gene, *atp8* (Pi = 0.264) presents the highest variability, followed by *nad2* (Pi = 0.240) and *nad6* (Pi = 0.238). However, *cox1* (Pi = 0.155) and *nad1* (Pi = 0.155) were the comparatively conserved genes in the 13 PCGs. Two rRNA genes were also highly conserved, with lower values of 0.136 in *rrnL* and *rrnS*, respectively.

### 3.6. Phylogenetic Analyses

Under the best models (Appendix A) selected by PartitionFinder and the heterogeneous model CAT + GTR, the ML and BI analyses based on PCG123R, PCG123, PCG12R and PCG12 datasets yielded similar tree topologies (Figure 7 and Appendix A), Bayesian posterior probabilities (PP) and bootstrap values (BS), all of which are shown on individual nodes. Overall, most received moderate to high support (Bayesian PP = 0.9–1/BS = 70–100) [35,36]. BI analysis and ML analysis provided better resolution with stronger support values. The results supported the monophyly of the lineages of the superfamily Fulgoroidea. The families Delphacidae, Achilidae, Ricaniidae, and Fulgoridae were also monophyletic in our analyses. Meanwhile, the results show that Fulgoroidea was divided into two groups: Delphacidae and ((Achilidae + (Lophopidae + (Issidae + (Flatidae + Ricaniidae)))) + Fulgoridae) with the PP = 1 and BS = 100, except the tree produced based on the PCG12R dataset under the CAT + GTR model, which was ((Lophopidae + (Issidae + (Flatidae + Ricaniidae))) + (Achilidae+ Fulgoridae)). The phylogenetic analysis also supported the early branching of the family Delphacidae with high support values (PP/BS = 1/100), supporting the previous studies of Urban and Cryan (2007) and Song and Liang (2013) [37,38]. Furthermore, the hypothetical relationships of (Lophopidae + (Issidae + (Flatidae + Ricaniidae))) in the superfamily Fulgoroidea were robustly supported. Within Fulgoridae, the phylogenetic relationships indicated that Aphaeninae was divided into two clades that we consider as representing the tribe Aphaenini Blanchard, 1847 (genera *Aphaena* Guérin-Méneville, 1834; *Limois* Stål, 1863; *Lycorma* Stål, 1863; *Penthicodes* Blanchard, 1845) and Pyropsini Urban and Cryan, 2009 (genera *Neoalcathous* Wang and Huang, 1989; *Pyrops* Spinola, 1839); *Dichoptera* sp. was branching off earlier than other species in Fulgoridae; the genus *Lycorma* was the sister to *Aphaena* (Figure 7); the tribe Limoisini is not recovered.

## 4. Discussion

### 4.1. Comparative Analysis of Fulgorid Mitogenomes

The nine sequenced fulgorid mitogenomes were conservative and similar to other species of Fulgoroidea from previous studies [33,39,40,41,42,43,44,45,46]. Their length ranged from 15,803 to 16,510 bp, comparable to the previously reported fulgorid mitogenomes that ranged from 15,946 bp (in *L. delicatula*) to 16,237 bp (in *A.* (*C.*) *amabilis*) [21,22,46]. These different lengths were mainly attributed to the varied size of intergenic spacer regions and the length of AT-rich regions.

The nine fulgorid mitogenomes exhibited an extremely high A + T content, ranging from 77.9% in *Dichoptera* sp. to 73.6% in *Py. spinolae*. These values are comparable to the previously reported fulgorid mitogenomes which ranged from 74.3% (in *Py. candelaria*) to 77.9% (in *A.* (*C.*) *amabilis*) [21,22,46]. The region at *nad3-nad4l* was extremely high in A + T content, serial tRNAs and stable stem-loop structures which may disrupt PCR and sequencing reactions; meanwhile, this study also found the poly (A) in *nad4* and *nad5*, presenting a high percentage of F residues. All these may cause the mitogenome sequences of most fulgorid species to become rather difficult. This phenomenon was also found in some other Hemiptera species [21,22,46,47,48,49,50].

Wang et al. (2019) suggested that a missing DHU stem in the *trnV* of *Aphaena* was unusual in Fulgoridae. Nevertheless, current evidence suggests that the mitogenomes of fulgorid species all lack stable DHU stems in *trnV* except *Dichoptera* sp. (Appendix A) [21,22,46], which made the missing DHU stem in *trnV* seem a common feature in Fulgoridae.

The size of the A + T-rich region in the nine sequenced species was similar to other Fulgoroidea, except *Sivaloka damnosus* Chow and Lu (in Issidae) [41]. Here, we found diversity in the repeat unit in Fulgoridae. The genus *Penthicodes* had a similar repeat unit as that found in *Aphaena* and *Lycorma*, and the same repeat unit was also found in *Pyrops* and *N. huangshanana*, but the repeat unit of *Dichoptera* sp. is remarkably different from the other sequenced species. Tandem repeats were also identified in the control region of the mitogenomes in several families in Fulgoroidea, including Ricaniidae, Flatidae, Delphacidae, Issidae, Fulgoridae and Achilidae [21,22,33,39,41,42,44,45]. The control region of Fulgoroidea presented a dramatic divergence in the base composition, fragment length and the repeat units.

### 4.2. Nucleotide Diversity of Fulgorid Mitogenomtes

Mitochondrial genes have been widely used as genetic markers, especially the *cox1* gene, including widespread use as DNA barcodes for identifying species and testing phylogenetic relationships among related taxa [51]. However, *cox1* remains inefficient for species delimitation in some groups owing to intra- and interspecific variations [52,53,54]. This analysis indicates that the *cox1* gene is relatively conserved and exhibits a slow evolution rate. The *nad1* gene is also highly conserved, while *atp8*, *nad2* and *nad6* genes have relatively faster evolution rates. Although *atp8* with 165 bp may be too short to be phylogenetically informative, *nad2* and *nad6* could be selected as potential DNA markers to clarify the phylogenetic relationships of closely related species of Fulgoridae.

### 4.3. Phylogeny and Species Delimitation

The genus *Pyrops* is endemic to the Indomalayan Region, which indubitably belongs to the Old World lineages, as indicated in the two alternative biogeographic hypotheses of Urban and Cryan (2009) [7]. Wang and Huang (1989) [55] erected the genus *Neoalcathous* and placed it in the subfamily Amyclinae based on its similarities to *Alcathous* Stål.

However, this placement of Oriental genera in a subfamily based on Neotropical taxa was shown to be erroneous by the subsequent molecular study of Urban and Cryan (2009) [7], and *Neoalcathous* was hence transferred to the subfamily Aphaeninae by Constant and Pham (2018) [17]. In the current study, *N. huangshanana* was placed in a clade with the genus *Pyrops*. The morphological characters also show similarities between *N. huangshanana* and the genus *Pyrops* in being larger; cephalic process curved dorsally; pronotum with median carina (sometimes obsolete) and a small but strongly impressed point on each side; mesonotum with median carina; tegmina colored, at most 3 times as long as broad, with the apical margin more or less rounded [17,55,56]. Here, we hypothesize that *N. huangshanana* is closely related to the genus *Pyrops* and we place them in a tribe Pyropsini Urban and Cryan, 2009, together with the closely related *Saiva* Distant, 1906, and *Datua* Schmidt, 1911 [7,57], in the subfamily Aphaeninae. The genus *Limois* apparently belongs to the tribe Aphaenini, and the tribe Limoisini might need to be considered a synonym of the latter in the future. However, further study including additional genera placed in Limoisini (*Bloeteanella* Lallemand, 1959, *Erilla* Distant, 1906, *Neolieftinckana* Lallemand, 1963, *Nisax* Fennah, 1977, *Ombro* Fennah, 1977, *Saramel* Fennah, 1977) is necessary to refine and confirm or refute the relevance of the tribe Limoisini.

*Dichoptera* sp. belongs to the tribe Dichopterini in the Dichopterinae. However, the assignment of the subfamily Dichopterinae was once controversial [13,14]. This study found *trnT* with a paired base in the aminoacyl stem and a lack of stable DHU stem in *trnV* in *Dichoptera* sp., which is obviously different from other fulgorid species. However, more evidence is still needed before ascertaining the status and placement of Dichopterinae, including the addition of representatives of Dictyopharidae into phylogenetic studies.

## 5. Conclusions

The nine fulgorid mitogenomes which belong to five genera (*Dichoptera* Spinola, 1839; *Neoalcathous* Wang and Huang, 1989; *Limois* Stål, 1863; *Penthicodes* Blanchard, 1840 and *Pyrops* Spinola, 1839) were sequenced. The arrangement of nine sequenced fulgorid mitogenomes are conservative and similar to other species of Fulgoridae. The comprehensive analysis can provide a more profound understanding of the mitochondrial characteristics among Fulgoridae.

The phylogenetic analyses showed that Delphacidae was the sister to ((Achilidae + (Lophopidae + (Issidae + (Flatidae + Ricaniidae)))) + Fulgoridae). The subfamily Aphaeni-nae was divided into Aphaenini and Pyropsini. The genus *Limois* is recovered in the Aphaeninae, apparently belongs to the tribe Aphaenini, and the Limoisini might need to be considered a synonym which needs further confirmation. *Dichoptera* sp. was the earliest branch in the Fulgoridae. Compared with other fulgorid mitogenomes, *Dichoptera* sp. has obvious difference in *trnT* and *trnV*. Ascertaining the status and placement of Dichopterinae still needed more study to do. Increasing the sample and mitochondrial data to solve the problems existing in the phylogeny of Fulgoridae is our future work.

## Figures and Tables

**Figure 1 genes-12-01185-f001:**
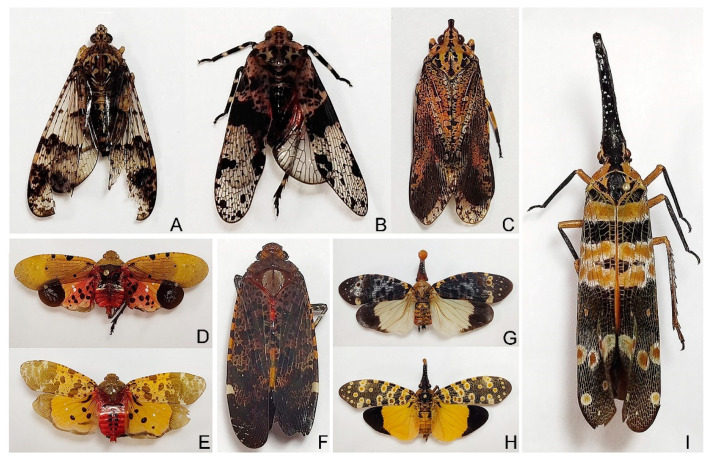
Photo plate of nine Fulgoridae specimens.

**Figure 2 genes-12-01185-f002:**
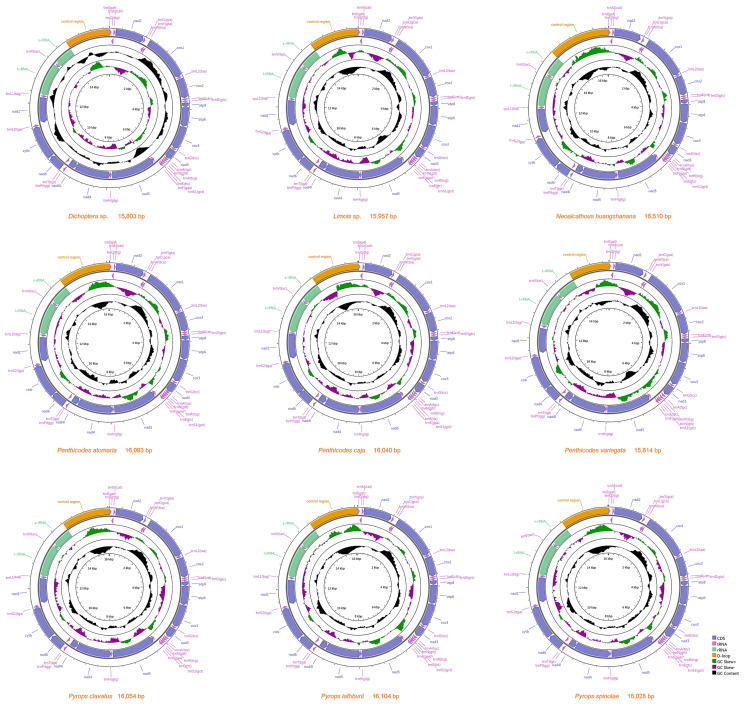
Organization of nine Fulgoridae complete mitogenomes.

**Figure 3 genes-12-01185-f003:**
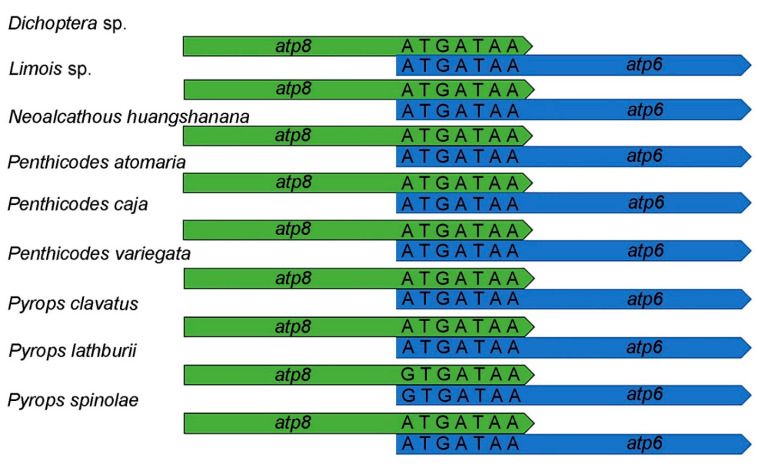
Sequence alignments of *atp8-atp6*.

**Figure 4 genes-12-01185-f004:**
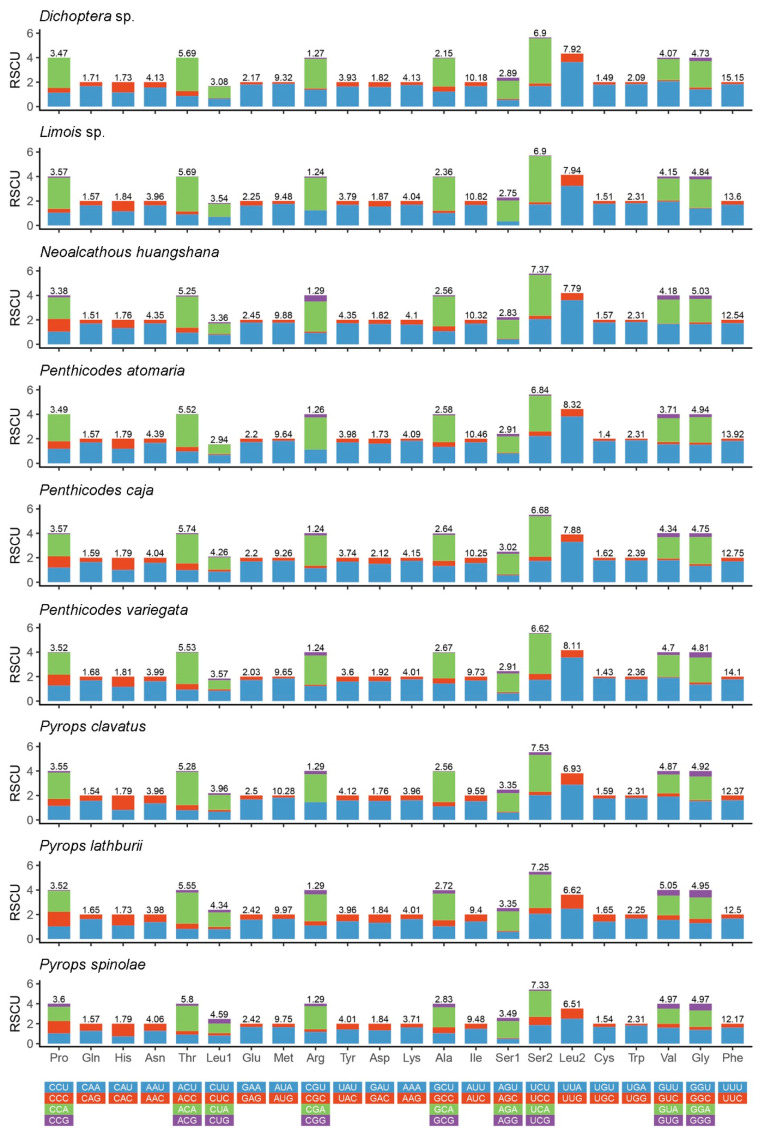
Relative synonymous codon usage (RSCU) in the mitogenomes of nine fulgorid species.

**Figure 5 genes-12-01185-f005:**
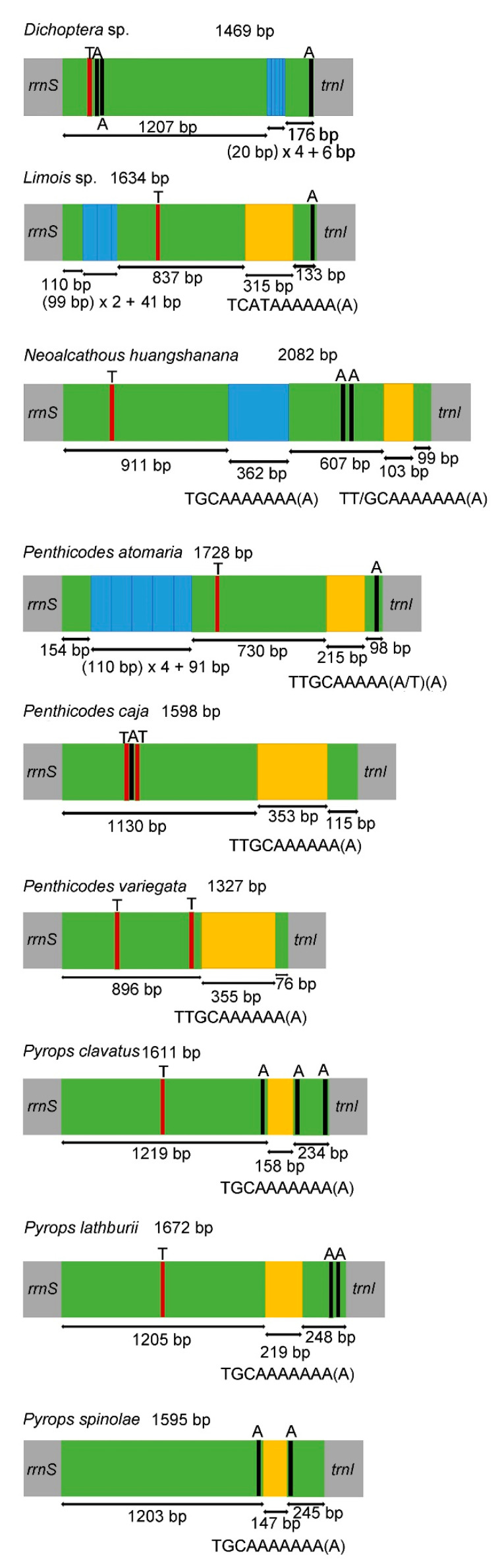
Structures of the control regions in the nine Fulgoridae mitogenomes. The blue and yellow boxes show the tandem repeats (the uppercase below represents the repeat units of sequences). The green boxes are non-repeat regions. The black and red blocks were the structures of poly (A) and poly (T), respectively.

**Figure 6 genes-12-01185-f006:**
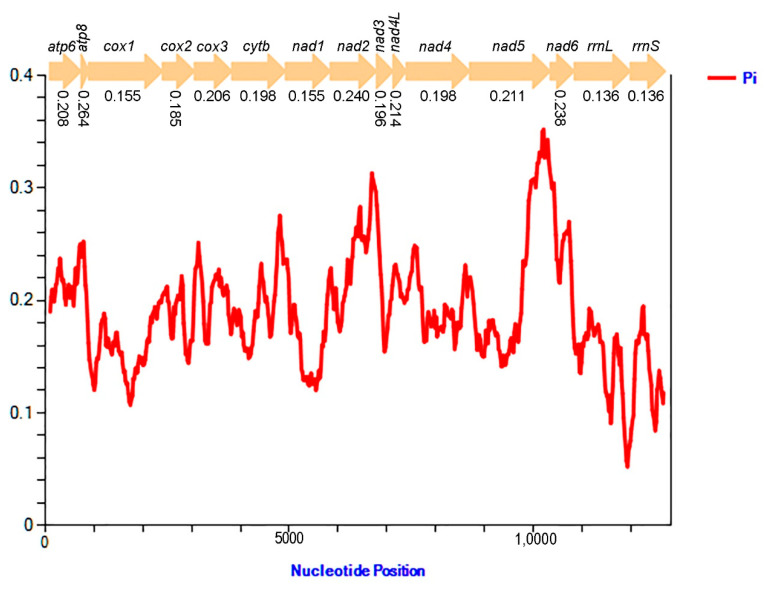
Sliding window analyses based on 13 PCGs and two rRNA genes of fourteen Fulgoridae mitogenomes. The red line indicates the value of nucleotide diversity (Pi) (window of 200 bp with the step size of 20 bp). The Pi value of each gene is below the gene name.

**Figure 7 genes-12-01185-f007:**
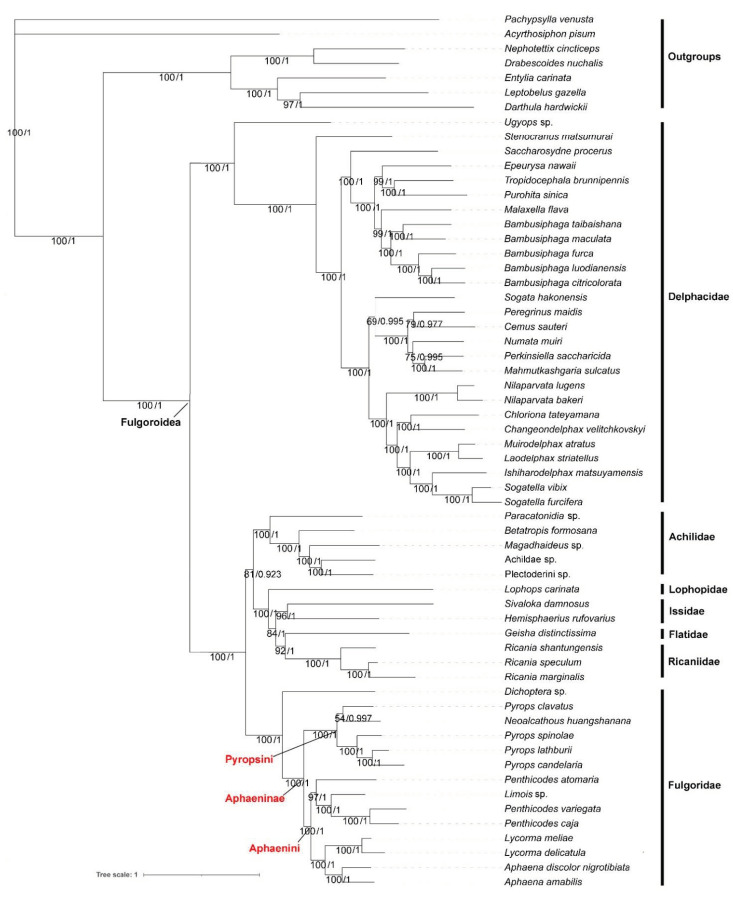
Phylogenetic tree inferred from ML and BI (MrBayes) analysis based on PCG123R matrix. Bootstrap values (BS) and Bayesian posterior probabilities (PP) are indicated on branches.

**Table 1 genes-12-01185-t001:** The mitogenomic sequences used in this study.

Superfamily	Family	Subfamily	Species	Accession Number
Fulgoroidea	Delphacidae	Delphacinae	*Nilaparvata lugens* (Stål)	JX880069
		Delphacinae	*Changeondelphax velitchkovskyi* (Melichar)	MG049916
		Delphacinae	*Peregrinus maidis* (Ashmead)	MG049917
		Delphacinae	*Nilaparvata bakeri* (Muir)	KC333655
		Delphacinae	*Sogatella furcifera* (Horváth)	KC512914
		Delphacinae	*Saccharosydne procerus* (Matsumura)	MG515237
		Delphacinae	*Bambusiphaga citricolorata* Huang and Tian	MH293452
		Delphacinae	*Bambusiphaga furca* Huang and Tian	MH293453
		Delphacinae	*Bambusiphaga luodianensis* Ding	MH293454
		Delphacinae	*Bambusiphaga maculate* Chen and Li	MH293455
		Delphacinae	*Bambusiphaga taibaishana* Qin	MH293456
		Delphacinae	*Epeurysa nawaii* Matsumura	MH293459
		Delphacinae	*Malaxella flava* Ding and Hu	MH293463
		Delphacinae	*Purohita sinica* Huang and Ding	MH293467
		Delphacinae	*Tropidocephala brunnipennis* Signoret	MH293471
		Delphacinae	*Cemus sauteri* (Muir)	MH293457
		Delphacinae	*Chloriona tateyamana* Matsumura	MH293458
		Delphacinae	*Ishiharodelphax matsuyamensis* (Ishihara)	MH293461
		Delphacinae	*Muirodelphax atratus* Vilbaste	MH293464
		Delphacinae	*Numata muiri* (Kirkaldy)	MH293465
		Delphacinae	*Perkinsiella saccharicida* Kirkaldy	MH293466
		Delphacinae	*Sogata hakonensis* (Matsumura)	MH293468
		Delphacinae	*Sogatella vibix* (Haupt)	MG515238
		Delphacinae	*Mahmutkashgaria sulcatus* (Ding)	MH293470
		Criomorphinae	*Laodelphax striatellus* (Fallén)	FJ360695
		Stenocraninae	*Stenocranus matsumurai* Metcalf	MH293469
		Asiracinae	*Ugyops* sp.	MH352481
	Lophopidae		*Lophops carinata* (Kirby)	MT990448
	Ricaniidae	Ricaniinae	*Ricania speculum* (Walker)	KX371891
	Ricaniidae	Ricaniinae	*Ricania marginalis* (Walker)	JN242415
		Ricaniinae	*Ricania shantungensis* (Chou and Lu)	NC_051496
	Flatidae	Flatinae	*Geisha distinctissima* (Walker)	FJ230961
	Issidae	Issinae	*Sivaloka damnosus* (Chou and Lu)	FJ360694
		Hemisphaeriinae	*Hemisphaerius rufovarius* Walker	MT210096
	Achilidae	Achilinae	*Betatropis formosana* Matsumura	MH324927
		Achilinae	*Magadhaideus* sp.	MH324928
			Achilidae sp.	MH324929
			Plectoderini sp.	MH324930
		Achilinae	*Paracatonidia* sp.	MH324931
	Fulgoridae	Aphaeninae	*Lycorma delicatula* (White)	EU909203
		Aphaeninae	*Lycorma meliae* Kato	MT079725
		Aphaeninae	*Aphaena discolor nigrotibiata* Schmidt	MN025523
		Aphaeninae	*Aphaena amabilis* (Hope)	MN025522
		Aphaeninae	*Penthicodes atomaria* (Weber)	MW662662
		Aphaeninae	*Penthicodes variegata* (Guérin-Méneville)	MW662664
		Aphaeninae	*Penthicodes caja* (Walker)	MW662663
		Aphaeninae	*Limois* sp.	MW662660
		Aphaeninae	*Neoalcathous huangshanana* Wang and Huang	MW662661
		Dichopterinae	*Dichoptera* sp.	MW662659
			*Pyrops candelaria* (Linné)	FJ006724
			*Pyrops clavatus* (Westwood)	MW662665
			*Pyrops lathburii* (Kirby)	MW662666
			*Pyrops spinolae* (Westwood)	MW662667
Outgroup	Membracidae		*Entylia carinata* (Forster)	NC_033539
			*Leptobelus gazella* Fairmaire	NC_023219
	Aetalionidae		*Darthula hardwickii* Gray,	NC_026699
	Cicadellidae		*Drabescoides nuchalis* (Jacobi)	NC_028154
			*Nephotettix cincticeps* (Uhler)	NC_026977
	Aphididae		*Acyrthosiphon pisum* (Harris)	NC_011594
	Aphalaridae		*Pachypsylla venusta* (Osten-Sacken)	NC_006157

**Table 2 genes-12-01185-t002:** Nucleotide composition of the mitogenomes of nine Fulgoridae.

Species	Regions	Size (bp)	T(U)	C	A	G	AT(%)	GC(%)	AT Skew	GC Skew
*D*.	PCGs	10,944	43.7	12.3	33.7	10.4	77.4	22.7	−0.13	−0.082
	1st codon position	3648	37.7	11.2	36.2	14.9	73.9	26.1	−0.02	0.141
	2nd codon position	3648	49.6	18.1	19.8	12.4	69.4	30.5	−0.429	−0.187
	3rd codon position	3648	43.8	7.4	45	3.9	88.8	11.3	0.014	−0.311
	tRNAs	1422	36.3	10.1	40.2	13.4	76.5	23.5	0.051	0.14
	rRNAs	1952	49.7	8	27.6	14.7	77.3	22.7	−0.287	0.296
	A + T-rich region	1469	40.1	9	43.6	7.4	83.7	16.4	0.041	−0.1
	Full genome	15,803	28.5	14.2	49.4	7.9	77.9	22.1	0.267	−0.283
*L*.	PCGs	10,956	42.5	12.5	33.9	11.2	76.4	23.7	−0.112	−0.056
	1st codon position	3652	36.3	11.7	36.6	15.4	72.9	27.1	0.005	0.136
	2nd codon position	3652	49.4	18.5	19.6	12.6	69	31.1	−0.432	−0.189
	3rd codon position	3652	41.8	7.3	45.5	5.5	87.3	12.8	0.042	−0.142
	tRNAs	1410	35.7	10.5	40.9	12.9	76.6	23.4	0.067	0.103
	rRNAs	1945	50	7.6	28.2	14.2	78.2	21.8	−0.278	0.302
	A + T-rich region	1634	34.4	9	51.2	5.4	85.6	14.4	0.196	−0.246
	Full genome	15,957	27.9	14.3	49.7	8	77.6	22.3	0.281	−0.282
*N*.	PCGs	10,929	42.2	12.5	33.8	11.5	76	24	−0.112	−0.044
	1st codon position	3643	36.1	11.3	36.6	16	72.7	27.3	0.006	0.173
	2nd codon position	3643	47.9	18.5	20.6	13	68.5	31.5	−0.398	−0.176
	3rd codon position	3643	42.7	7.8	44.1	5.4	86.8	13.2	0.016	−0.178
	tRNAs	1410	35.7	11.2	39.9	13.3	75.6	24.5	0.055	0.084
	rRNAs	1945	48.6	8.7	28.2	14.6	76.8	23.3	−0.266	0.252
	A + T-rich region	2082	28.7	7.9	56	7.4	84.7	15.3	0.322	−0.028
	Full genome	16,510	28.6	13.4	48.7	9.3	77.3	22.7	0.261	−0.18
*Pea*.	PCGs	10,959	43.5	12.2	33.7	10.7	77.2	22.9	−0.127	−0.066
	1st codon position	3653	37	11	36.9	15.1	73.9	26.1	−0.001	0.157
	2nd codon position	3653	48.8	18.4	20	12.8	68.8	31.2	−0.419	−0.179
	3rd codon position	3653	44.6	7.1	44.2	4.1	88.8	11.2	−0.005	−0.273
	tRNAs	1399	36.5	10.4	40.2	12.9	76.7	23.3	0.048	0.104
	rRNAs	1945	48.8	7.9	28.3	15	77.1	22.9	−0.265	0.312
	A + T-rich region	1728	34.8	10.2	48.3	6.8	83.1	17	0.162	−0.201
	Full genome	16,093	29.2	13.9	48.6	8.3	77.8	22.2	0.249	−0.254
*Pec*.	PCGs	10,953	41.6	13.7	33.4	11.4	75	25.1	−0.11	−0.092
	1st codon position	3651	35.3	12.4	36.3	16	71.6	28.4	0.015	0.126
	2nd codon position	3651	48.6	18.6	19.9	13	68.5	31.6	−0.42	−0.177
	3rd codon position	3651	40.9	10.1	43.9	5.1	84.8	15.2	0.035	−0.324
	tRNAs	1428	36.1	10.9	39.5	13.5	75.6	24.4	0.045	0.106
	rRNAs	1977	48.9	7.9	28.2	15	77.1	22.9	−0.269	0.307
	A + T-rich region	1598	44.6	9.8	40.7	5	85.3	14.8	−0.046	−0.322
	Full genome	16,040	29.2	15.5	47.3	8.1	76.5	23.6	0.237	−0.314
*Pev*.	PCGs	10,947	43.1	12.8	33	11.1	76.1	23.9	−0.132	−0.071
	1st codon position	3649	36.4	11.8	35.7	16.1	72.1	27.9	−0.01	0.154
	2nd codon position	3649	49.7	18.3	19.3	12.7	69	31	−0.441	−0.179
	3rd codon position	3649	43	8.4	44	4.6	87	13	0.011	−0.294
	tRNAs	1422	35.9	11	39.5	13.6	75.4	24.6	0.048	0.105
	rRNAs	1960	48.4	8	28.8	14.8	77.2	22.8	−0.254	0.298
	A + T-rich region	1327	36.9	12.7	43.5	6.8	80.4	19.5	0.082	−0.305
	Full genome	15,814	28.6	15	48.1	8.4	76.7	23.4	0.254	−0.282
*Pyc*.	PCGs	10,950	40.7	13.8	33.4	12.1	74.1	25.9	−0.099	−0.065
	1st codon position	3650	35.1	12.1	36.3	16.5	71.4	28.6	0.017	0.156
	2nd codon position	3650	47.8	18.8	19.9	13.4	67.7	32.2	−0.413	−0.168
	3rd codon position	3650	39.1	10.5	43.9	6.4	83	16.9	0.057	−0.242
	tRNAs	1405	35.1	11.9	38.9	14.2	74	26.1	0.051	0.087
	rRNAs	1948	48.7	8.8	27.6	14.9	76.3	23.7	−0.277	0.257
	A + T-rich region	1611	32.3	8.5	51.9	7.3	84.2	15.8	0.232	−0.079
	Full genome	16,054	26.7	15.3	48.6	9.4	75.3	24.7	0.29	−0.24
*Pyl.*	PCGs	10,956	40.1	14.7	32.3	13	72.3	27.7	−0.108	−0.059
	1st codon position	3652	34.4	12.5	36.1	16.9	70.5	29.4	0.024	0.151
	2nd codon position	3652	47.7	19	19.9	13.4	67.6	32.4	−0.412	−0.171
	3rd codon position	3652	38	12.6	40.7	8.7	78.7	21.3	0.034	−0.18
	tRNAs	1421	34.8	11.4	39.9	13.9	74.7	25.3	0.069	0.1
	rRNAs	1943	48.5	9.6	26.5	15.4	75	25	−0.294	0.232
	A + T-rich region	1672	32.2	8.7	51.3	7.8	83.5	16.5	0.228	−0.051
	Full genome	16,104	26.4	15.5	47.7	10.4	74.1	25.9	0.288	−0.199
*Pys.*	PCGs	10,959	39.6	15.3	31.8	13.2	71.4	28.5	−0.109	−0.073
	1st codon position	3653	34.1	12.8	36.2	17	70.3	29.8	0.03	0.141
	2nd codon position	3653	47.3	19.5	19.7	13.6	67	33.1	−0.413	−0.18
	3rd codon position	3653	37.4	13.7	39.7	9.2	77.1	22.9	0.03	−0.197
	tRNAs	1420	34.7	11.1	39.7	14.5	74.4	25.6	0.067	0.135
	rRNAs	1944	48.3	9.6	27	15.1	75.3	24.7	−0.284	0.222
	A + T-rich region	1595	32.8	8.8	50.8	7.6	83.6	16.4	0.215	−0.069
	Full genome	16,028	26.4	15.8	47.2	10.7	73.6	26.5	0.283	−0.193

*Dichoptera* sp. (*D*.); *Limois* sp. (*L*.); *Neoalcathous huangshanana* (*N*.); *Penthicodes atomaria* (*Pea*.); *Penthicodes caja* (*Pec*.); *Penthicodes variegata* (*Pev*.); *Pyrops clavatus* (*Pyc*.); *Pyrops lathburii* (*Pyl*.) and *Pyrops spinolae* (*Pys*.).

## Data Availability

Not applicable.

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
