# Peer review of "Characterization, Comparative Analysis and Phylogenetic Implications of Mitogenomes of Fulgoridae (Hemiptera: Fulgoromorpha)"

_genes, 2021, doi:10.3390/genes12081185_

Round 1

Reviewer 1 Report

This work presents original data on the mitochondrial genomes of multiple Old World Fulgoridae species that is important and relevant, and will be of great use in a number of further studies. The quality of the genomics and analytic aspects of this work are very strong. The phylogenetic results presented here, with respect to relationships among the families within Fulgoroidea, are somewhat limited and further work is necessary to more rigorously reconstruct these relationships; i.e., that will greater taxonomic representation, particularly of New World Fulgoridae as well as Dictyopharidae. But the results found in the phylogenetic reconstructions presented here are interesting and useful. The need for further sampling of Dictyopharidae was mentioned in the manuscript lines 375-380, which was great to see, particularly because of the issues surrounding placement of Dichopterinae in Fulgoridae versus Dictyopharidae. Overall, this is a useful contribution.

Author Response

Reviewer 1:

This work presents original data on the mitochondrial genomes of multiple Old World Fulgoridae species that is important and relevant, and will be of great use in a number of further studies. The quality of the genomics and analytic aspects of this work are very strong. The phylogenetic results presented here, with respect to relationships among the families within Fulgoroidea, are somewhat limited and further work is necessary to more rigorously reconstruct these relationships; i.e., that will greater taxonomic representation, particularly of New World Fulgoridae as well as Dictyopharidae. But the results found in the phylogenetic reconstructions presented here are interesting and useful. The need for further sampling of Dictyopharidae was mentioned in the manuscript lines 375-380, which was great to see, particularly because of the issues surrounding placement of Dichopterinae in Fulgoridae versus Dictyopharidae. Overall, this is a useful contribution.

Answer: Thank you very much for your approval.

Reviewer 2 Report

The paper by Wang and collaborators investigated the phylogenetic relationships of a group of Hemiptera, the Fulgoromorpha by sequencing denovo and characterize nine mitogenomes and by comparing these new mitogenomes with already published mitogenomes of this group of Hemiptera.  

The paper is nice but the authors should address some problems before in can be accepted for publication The manuscript need also a language revision. On this last point, it look like that the coauthor did not read the last submitted version of the ms.  For example none of the coauthors is affiliated to the Department of Entomology, University of Delaware.   

My major concerns on the manuscript are: The authors decided to use PCGs and rRNA for their phylogenetic analysis and exclude the tRNA without offering any explanation for this exclusion.

The authors applied partition finder to find the partition and best model to apply to each of them. But did they check for compositional difference among species. Furthermore partition finder as any other best model search software search among a set of models and found the best among them. This does not mean that the model is adequate. The authors should check for model adequacy for their partition (phylobayes, beast can do this).

One of the outgroups in the trees is Pachypsylla venusta whereas  in Table 1 there is Philaenus spumarius (Linnaeus). Which one has been used and is correct?

Node supports in the Bayesian inferences are posterior probabilities and not bootstraps (figure 8). Considering that ML and BI have identical topology, either one of the two could be moved to supplementary materials.

One of the coauthors published a similar paper on another groups of Fulgoroidea using three mitogenomes of Flatidae. This paper is not cited in the ms, Why? And more importantly why the three mitogenomes were not included in this work which would have allowed increasing taxa sampling for the Flatidea in the phylogenetic analysis. (Characterization of Three Complete Mitogenomes of Flatidae (Hemiptera: Fulgoroidea) and Compositional Heterogeneity Analysis in the Planthoppers' Mitochondrial Phylogenomics by Deqiang Ai  1 , Lingfei Peng  2 , Daozheng Qin  1 and  Yalin Zhang 2021).

A conclusion on the significance of this work is needed at the end of the discussion.

Minor points:

Tables 2, 3 and 4 can be probably moved to the supplementary materials.

Line 1 Article

Line 5 and 11 affiliation to correct

Line 23 with either the standard start codon of ATN or  the nonstandard GTG

Line 36 space before and

Line 57 space before are

Line 91-93 Geneious did not chose as default parameters the mitogenomes of L. delicatula etc.

Line 119 MAFFT

Line 157 What is compared to the nine mitogenomes?

Line 169 and 170  of the nine mitogenomes/species

Line 171 position was much higher

Line 172 had a positive … all nine species

Line 174 delete , such as ATT… ATC,

Line 184 and in 185 the least used

Line 186 delete are

Line 195 delete also

Line 202 were

Line 269. Are the conserved gene. Revise this sentence. Also the sentence 271-272 does not make sense.

Line 286 Support the early branching of the of the family Delphacidae with high support values.  

Line 292, 293 and maybe elsewhere. You need to put the name of who describe/names a species, not any other superior taxonomic ranking as tribe

Line 304 These

Line 305 delete in this study

Line 305 the.. ranged

Line 308 high in A+T

Line 308 - 312 I do not understand these sentences

Line 311 as(?) high

Line 360 -361 ?? or less rounded at most 3 times as long as broad ??

Round 2

Reviewer 2 Report

The paper by Wang and collaborators has improved from an analytical point of view after the first round of revision, however still remain language corrections that need to be addressed. These that follow are just some examples. The paper would benefit from a language revision as suggest after the first round.

Line 100 were chosen

Line 101-103 Geneious 11.0.2 was used for mitogenomes annotation with … used as references.

Line 137 two simultaneous runs of four chains each.

Line 138 settings of 5x106

Line 141 of the four

Line 144 delete After

Line 145 burn-in and a consensus

Line 178 The total size of the 13 PCGs of the none mitogenomes ranged between 10929 bp and 10959 bp. In all nine mitogenomes PCGs showed negative AT and GC skews.

Line 180 than of the first

Line 309 The nine fulgorid mitogenomes exhibited an extremely high A+T content, ranging from 77.9% in Dichoptera sp. to 73.6% in Py. Spinolae. These values are comparable to … 
